# The Application of Adaptive Tolerance and Serialized Facial Feature Extraction to Automatic Attendance Systems

Chun-Ling Lin * and Yi-Huai Huang

Department of Electrical Engineering, Ming Chi University of Technology, No. 84, Taishan Dist.,
New Taipei City 243, Taiwan; m07128014@mail2.mcut.edu.tw
* Correspondence: ginnylin@mail.mcut.edu.tw; Tel.: +886-2-2908-9899 (ext. 4819)

**Abstract:** The aim of this study was to develop a real-time automatic attendance system (AAS) based on Internet of Things (IoT) technology and facial recognition. A Raspberry Pi camera built into a Raspberry Pi 3B is used to transfer facial images to a cloud server. Face detection and recognition libraries are implemented on this cloud server, which thus can handle all the processes involved with the automatic recording of student attendance. In addition, this study proposes the application of data serialization processing and adaptive tolerance vis-à-vis Euclidean distance. The facial features encountered are processed using data serialization before they are saved in the SQLite database; such serialized data can easily be written and then read back from the database. When examining the differences between the facial features already stored in the SQLite databases and any new facial features, the proposed adaptive tolerance system can improve the performance of the facial recognition method applying Euclidean distance. The results of this study show that the proposed AAS can recognize multiple faces and so record attendance automatically. The AAS proposed in this study can assist in the detection of students who attempt to skip classes without the knowledge of their teachers. The problem of students being unintentionally marked present, though absent, and the problem of proxies is also resolved.

**Keywords:** automatic attendance system (AAS); Internet of Things (IoT); face detection; face recognition; Euclidean distance; SQLite





## 1. Introduction

The regular attendance of students is a prerequisite for good academic performance. Student attendance is one of the most important issues for all educational institutions. There are two common types of student attendance systems in use: manual attendance systems (MSA) and automatic attendance systems (AAS) [1]. The term MSA covers the traditional systems that require teachers to fill in attendance sheets manually. Where classes are larger, this can be difficult to administer because teachers often face enormous pressures, and it takes a lot of time to collect the necessary details about each student's name and record these without any errors. However, of late, information technology [2–5] has been widely used to provide convenience, speed up the task, and also streamline it. For the purposes of AAS, various technologies, such as facial recognition, iris detection, and radio frequency identification (RFID), have been used [6]. RFID systems utilize sensors in order to read data. The usefulness of RFID for this application lies in the fact that it can facilitate lecturers and, indeed, students with monitoring class attendance, via an AAS [7]. On the other hand, there are also some disadvantages, such as the fact that RFID is not as secure, again for this application, as biometric methods; the system is prone to manipulation [8]. That is, another person can use same RFID to do the roll call. In recent years, systems have been provided with powerful tools supporting artificial intelligence (AI) operations, including machine learning operations based on high bandwidth CPUs, GPUs, and specific AI accelerators. One such operation, facial recognition [9], has attracted a great deal of research attention and

has been subject to sustained development over the past 30 years. It has great potential for use across numerous government and commercial applications [10–12]. Security cameras with facial recognition capabilities are presently common in airports, offices, universities, ATM installations, banks, and indeed, in any location with a security system [13–15]. Parmar and Mehta described the most common methods of face recognition, such as the holistic matching method, the feature extraction method, and hybrid methods; and proposed a number of face recognition applications, such as face identification, access control, and security and identity verification [13]. Khandelwal et al. suggested various face recognition processing methods and adopted a Convolutional Neural Network (CNN) to develop a face recognition algorithm [14]. Norouzi explored face recognition based on deep neural networks and described various facial expression recognition applications [15], such as student attendance systems and building security systems.

Among all the techniques applied to AAS, facial recognition is considered to be the most efficient [16]. Although facial recognition based on deep learning techniques [17–19] has high accuracy, the model is complex and the recognition speed is slow, even when working on just a single image. In order to realize real-time facial recognition of students, two particular computer vision libraries—the open-source computer vision libraries (OpenCV) [20,21] and Dlib [22–24]—have been applied in many previous studies. Kar et al. adopted the OpenCV and Light Tool Kit (FLTK) to develop an AAS system and suggested that OpenCV can provide a simple-to-use computer vision infrastructure that can help people build fairly sophisticated vision applications quickly. The OpenCV library contains over 500 functions that span many areas of vision [20]. Joseph et al. adopted face detection, 128-dimension face encoding extraction, and support-vector machine (SVM) training to develop a system [21]. Boyko et al., explored the features and analyzed the pros and cons of OpenCV and Dlib [22]. Xu et al., adopted Dlib to detect faces, and then adopted CNN and a deep residual network (ResNet) for real-time face recognition [23]. Ambre et al. explored a real-time face recognition system using easily attainable components and libraries, such as Raspberry PI; Dlib, a Face Recognition library; and OpenCV [24].

This has been done on the basis that the utilization of libraries allows easy application and reimplementation. The OpenCV library is a cross-platform computer library which can provide an infrastructure for computer vision applications and allow the use of machine perception easily within commercial products. However, OpenCV [25] suffers from the problems of missed detections, false detections, and in effect poor recognition overall. On the other hand, Dlib is a cutting-edge toolbox which includes various different AI tools and which allows the creation of complex software programs for solving real-world problems [26]. Dlib can deal with bad and inconsistent lighting and various facial positions (such as tilted or rotated). It also achieves high performance within real-world implementations both in terms of speed and accuracy [27]; all this makes it suitable as a basis from which to develop an AAS. The challenges of AAS include how to make the attendance registration and management systems efficient, time-saving, simple, and easy. Furthermore, an AAS needs to recognize, accurately, multiple faces, faces in inconsistent lighting conditions, and faces presented via various positions. Pandey et al. combined the on-demand resource availability of cloud computing, which can store and retrieve the captured video anytime. Additionally, its surveillance mechanism involves the Viola–Jones algorithm for face detection by analyzing captured data [28]. Shanthi and Svalakshmi developed a surveillance method by using a face recognition based unmanned aerial vehicle (UAV) [29]. The method of UAV was implemented on Raspberry Pi module with Python libraries which included OpenCV, Dlib, Face_recognition, and Numpy. Gupta and Singh used in-memory computation to develop a real-time face recognition system. Even while the number of faces is increasing, it can still keep the frame rate steady during the entire process [30]. These recent studies developed an AAS system based on Raspberry Pi, a cloud computing environment, and face detection algorithms. They can make the systems simply, easily, and more efficiently.

The aim of this study was to develop a real-time AAS system based on Internet of Things (IoT) technology and facial recognition, which can handle the registration of students. The main contributions of this study are threefold:

1. Building a cloud server to develop face detection and recognition algorithms. A Raspberry Pi Camera system built into a Raspberry Pi 3B was used to capture and transfer images to a cloud server with a high-speed GPU. The development of our face detection and recognition algorithms was based on this cloud server, which handles the processes involved in registering student attendance, namely, comparing faces using extracted features stored in a database.

2. Creating an SQLite database to save students' facial features based on data serialization, which we propose as the strategy for this process. In order to create the SQLite database, the students' facial features will be put through data serialization before they are saved using SQLite.

3. Improving the performance of face recognition using adaptive tolerance with respect to Euclidean distance. In comparing the facial features found in the database with the new facial features encountered in the input, the former needs to be subjected to data deserialization. The proposed adaptive tolerance method can improve the performance of the facial recognition processes using Euclidean distance.

With the implementation of our proposed AAS, students cannot skip classes without the knowledge of their teachers.

This study is organized as follows. In Section 2, the proposed AAS schematic and methods are described. Section 3 shows the results of this study. Section 4 discusses the proposed AAS system and indicates the advantages of the proposed methods. Section 5 provides concluding remarks.

## 2. Methods

Figure 1 shows the schematic of our proposed AAS. The design of our AAS system includes four main components: a Raspberry Pi camera built into a Raspberry Pi 3B; a Django background processing system, the OpenCV image processing library, and the Dlib face detection and face recognition module. For each student, the proposed AAS system must save a headshot, and a complete set of facial features derived from this, onto the cloud server. The dimensions of the headshot are reduced to a standard size and then converted to a grayscale image, using OpenCV. Once this is done, the facial landmarks are identified using the Dlib library. Each set of facial landmarks derived from the students' faces is saved in SQLite [31] and associated with an appropriate students' names. A Raspberry Pi camera built into a Raspberry Pi 3B was set up in the classroom in order to capture the images and transfer them to a cloud server through the RESTful application programming interface (API). When the cloud server receives an image, the facial landmarks from this can be extracted using the Dlib library. The newly extracted features can then be compared with those already stored in the SQLite database in order to confirm (or not) the identity of the student. If this identity verification is successful, the student is recognized and so the student's name will show on a computer screen in the classroom as having been marked as in attendance.

### 2.1. The Cloud Server

The Django cloud server has an automatic background task management interface and can provide technical support systems, including a user authentication mechanism; it (the server) is used to receive the facial images, store these in the database, and process them [32]. This study adopted the Django system as the basis for implementing Raspberry Pi as a web server [33], applying the RESTful API [34]. The latter is an architectural style API that uses HTTP requests to access and use data. Into the Django system, we imported the OpenCV and Dlib libraries. These support the engine in processing the images and so detecting and recognizing the faces of the students.

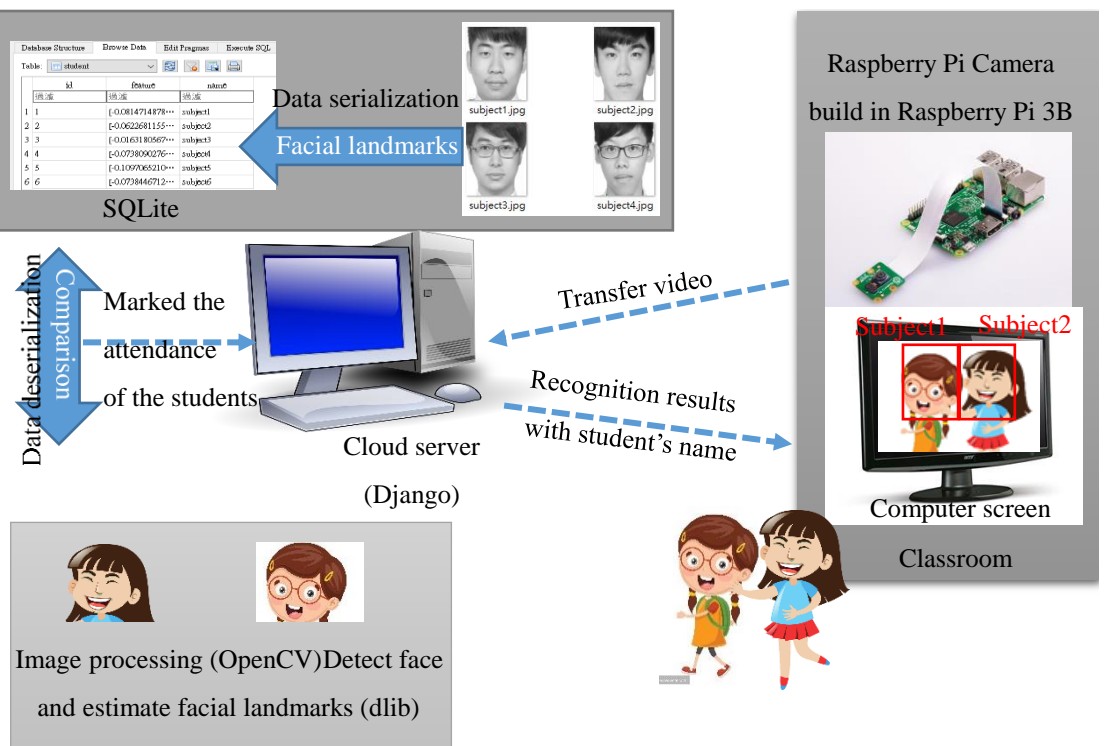

**Figure 1.** AAS schematic proposed in this study. There are four main components: a Raspberry Pi camera built into a Raspberry Pi 3B, a Django background processing system, the OpenCV image processing library, and the Dlib face detection and face recognition module. In the SQLite, the meaning of 過濾 is filter.

*2.2. SQLite Databases*

The flowchart in Figure 2 shows how the student database is created via SQLite. SQLite is a popular database management system [35]. In this case, the SQLite database will contain information regarding the students to be detected and recognized. Each student, on enrolment, must provide a headshot which presents a complete set of facial features, as shown in Figure 3a. The convolutional neural network's (CNN) features, used within the context of the Maximum-Margin Object Detector (MMOD) (MMOD-CNN) face detector in the Dlib library (Dlib CNN), are adopted as the basis by which the faces and their positions within the object are detected, identified, and recognized [36–38]. The face within the object is extracted from the original headshot and converted to grayscale using OpenCV (Figure 3b). For the Dlib facial recognition network, the image being processed must be transformed into an output feature vector with 128 numerical facial features. These are used to characterize each face [39], as shown in Figure 4.

In order to store, in SQLite, the output feature vectors, each representing the 128 numerical features which in turn represent a particular face, the feature vector needs to be subjected to data serialization. This technique converts data objects represented as complex data structures into a byte stream for storage, transfer, and distribution purposes on physical devices, using JavaScript Object Notation (JSON library) [40]. The name of the subject and the 128 numerical facial features, after JSON serialization, will be stored in the SQLite database, as shown in Figure 5.

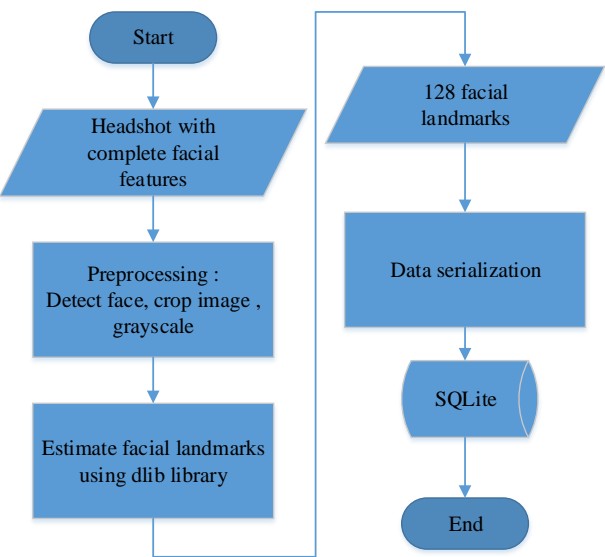

**Figure 2.** Flowchart showing the process of creating the student database in SQLite.

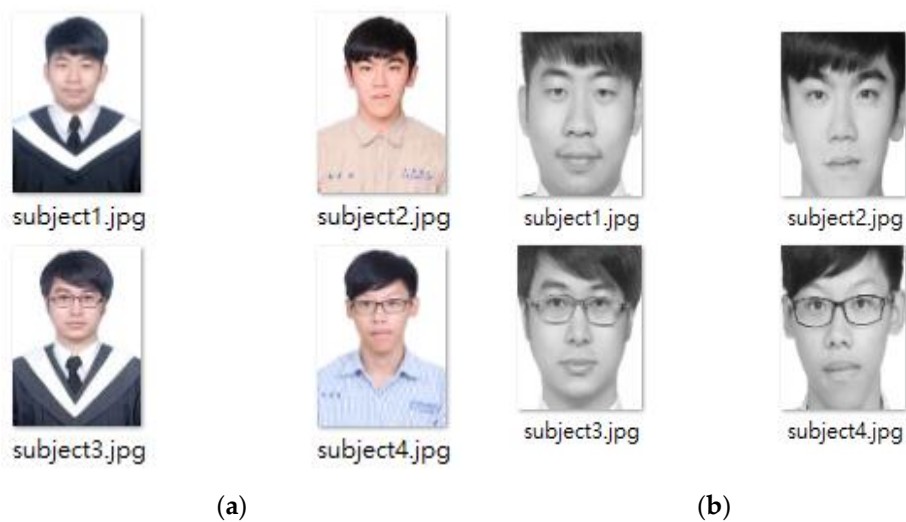

(**a**)                                       (**b**)

**Figure 3.** The headshot with complete facial feature. (**a**) The headshot which presents a complete set of facial features. (**b**) The face within the object is extracted from the original headshot and converted to grayscale using OpenCV.

**Figure 4.** Output feature vector with128 numerical facial features by Dlib facial recognition network which was sipped from the python results.

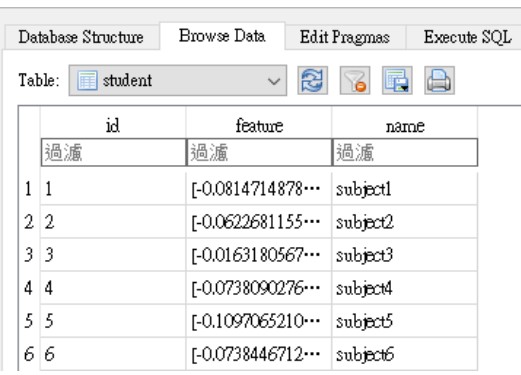

**Figure 5.** SQLite databases after JSON serialization which was sipped from the SQLite software results The meaning of 過濾 is filter.

*2.3. Face Recognition*

Figure 6 shows the flowchart of the face recognition process proposed and described as follows:

1.  The real-time continuous face recognition AAS, which captures the attendance and duration of attendance of students from the Raspberry Pi camera set-up, requires that Raspberry Pi Model B.
2.  The raspberry Pi model extracts an image from every six frames of the live video (of the classroom) and uploads them to the cloud server.
3.  The MMOD-CNN process has been adopted as the process by which the faces—and their positions within the object—are detected. The object is a list of rectangular objects. If the MMOD-CNN cannot obtain rectangle objects, we go back to Step 2.
4.  OpenCV crops the image of each face with the object and converts it to grayscale form.
5.  The estimated 128 facial landmarks are obtained using the Dlib facial recognition network.
6.  For facial similarity calculation, the estimated facial landmarks are compared with the facial landmarks currently stored in the SQLite database. This comparison is performed after the JSON deserialization process has been applied to the existing facial features stored in the SQLite database, using Euclidean distance [41] as follows:

$$\sum_{i=1}^{128} \sqrt{(x_{1i} - x_{2i})^2} \tag{1}$$

where $x_1$ is one of the facial landmarks which currently exists in the SQLite database, and $x_2$ is a just-estimated facial landmark derived from a new image of a student.

7.  If the value of the Euclidean distance is smaller than the tolerance, this means that the student whose image is being processed has been identified.
8.  The name of the student can be shown on the computer screen, and the student is marked as present via the cloud server.

2.3.1. Maximum-Margin Object Detector Convolutional Neural Network (MMOD-CNN)

Dlib has two types of face detection tool (https://pyimagesearch.com/2021/04/19/face-detection-with-Dlib-hog-and-cnn/ (accessed data:15 June 2022)): one is based on a classic histogram of oriented gradient (HOG) features, in conjunction with the linear classifier SVM, pyramid images, and a sliding window detection scheme; the other is based on the Maximum-Margin Object Detector convolutional neural network (MMOD-CNN), a face detector that uses deep learning face detection. The MMOD-CNN can be adopted using cnn_face_detection.py; the function returns a list of rectangle objects which extract the face within the object.

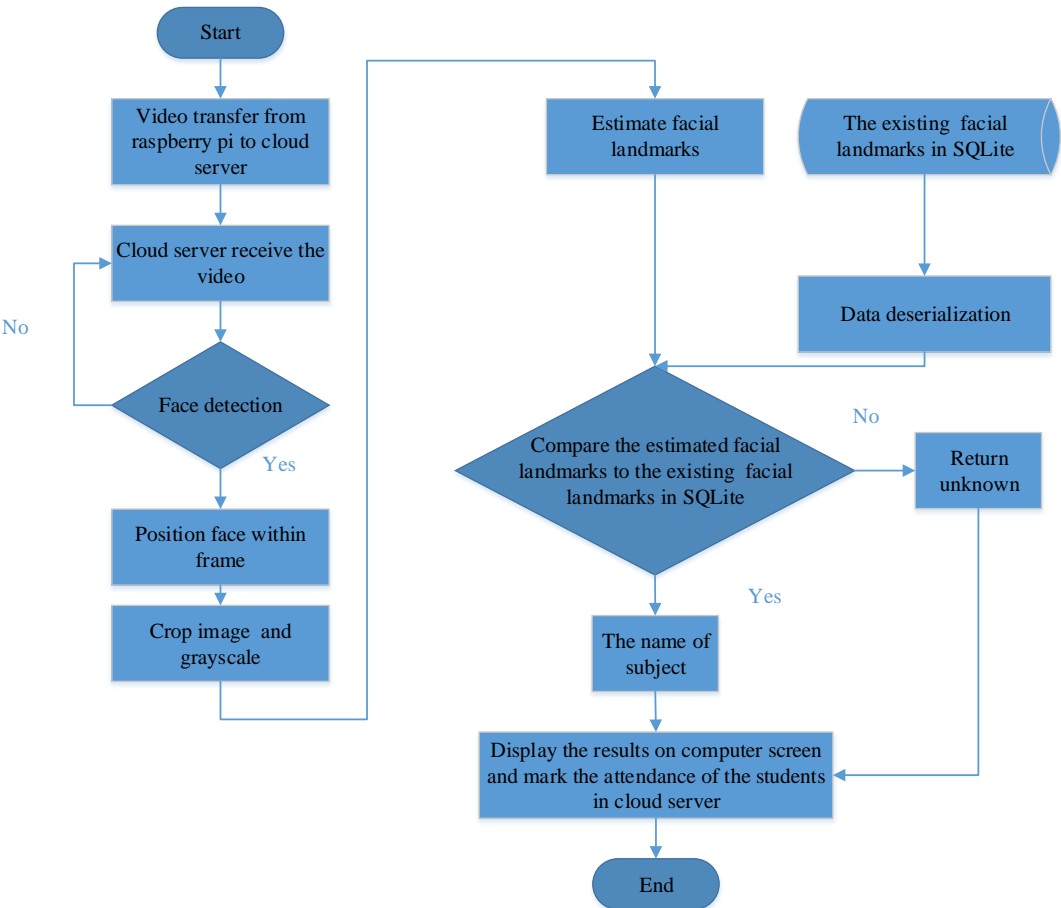

**Figure 6.** Flowchart of face recognition process.

### 2.3.2. Adaptive Tolerance

The definition of the tolerance, in terms of Euclidean distance, is a challenging problem in terms of face detection. In order to find a suitable tolerance, we set up a camera at a classroom door and asked six subjects to stand two meters from that door with their faces uncovered. With regard to these subjects, the smallest values of Euclidean distance between estimated facial landmarks and existing (in the database) facial landmarks can be observed in Table 1. Thus, in this case, the tolerance in terms of Euclidean distance can be defined as 0.44.

**Table 1.** Euclidean distances from six subjects.

| Subject | Euclidean Distance |
|---|---|
| Subject1 | 0.458 |
| Subject2 | 0.433 |
| Subject3 | 0.427 |
| Subject4 | 0.417 |
| Subject5 | 0.44 |
| Subject6 | 0.451 |
| **Average** | **0.438** |

However, in general, tolerance has to be defined as a fixed number, whereas the performance of facial recognition will be variable because, for instance, the distance between the student and the camera will affect this performance, as shown in Figure 7. When a

student is a significant distance from the camera, the face she or he presents will be relatively small. Although the MMOD-CNN process can, nevertheless, detect the face and find its position within the object, the number of pixels representing the face becomes very limited, which means that the detailed facial landmarks cannot be extracted via the Dlib facial recognition network. This causes the Euclidean distances involved to increase overall. As a result, students cannot necessarily be identified, as shown in Figure 7b.

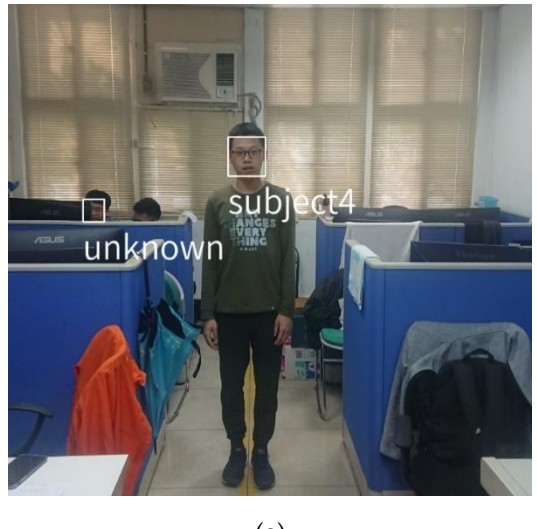
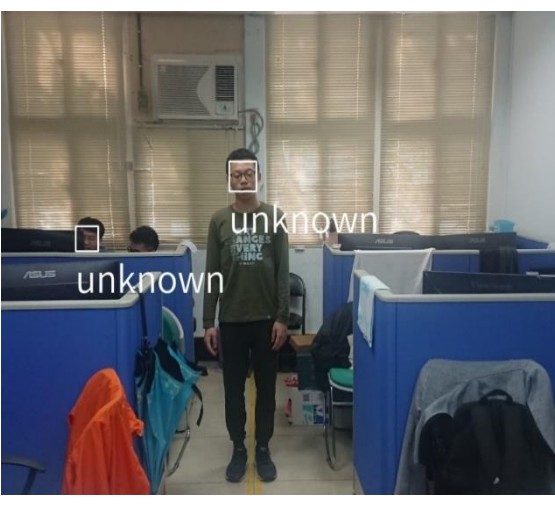

(**a**)                                          (**b**)

**Figure 7.** The performance of facial recognition, keeping the tolerances the same throughout (0.44). (**a**) The middle subject in the image is 0.68 cm and can be recognized. (**b**) The middle subject in the image is 0.47 cm and cannot be recognized.

Table 2 shows the values of the Euclidean distances between some estimated facial landmarks and some existing facial landmarks (existing, as in being present within the database), when the widths of the faces involved are different; see Figure 7. In Figure 7b, the width of the face of the middle subject in the image is 0.47 cm, and that of the left subject in the image is 0.39 cm. The smallest value of Euclidean distance, as between the middle subject and the existing dataset, is that between this subject and subject 4. If the tolerance with respect to the Euclidean distance is equal to 0.44, the facial recognition process will yield the value "unknown." This study proposes a method whereby an adaptive tolerance can be defined, based on the width of the face presented. Table 2 shows that the width of the face represented by it is equal to 0.68, and this can be recognized accurately using the tolerance with respect to Euclidean distance of 0.44. Thus, the standard width of face is defined as 0.7 cm, and the standard tolerance with respect to Euclidean distance as 0.44. When the width of the face is larger or smaller than 0.7, we say that the definition of the tolerance will vary as follows:

$$\text{Adaptive Tolernace} = 0.44 \times \left( \frac{0.7}{10} + \text{WF} \right) / \text{WF} \tag{2}$$

where WF is the width of the face presented.

**Table 2.** The values of the Euclidean distance with respect to two subjects when the sizes of faces are different, as in Figure 7.

| Width of Face = 0.39 cm (Left Subject) | Euclidean Distance in Figure 7a |
|---|---|
| Subject 1 | 0.5833 |
| Subject 2 | 0.6239 |
| Subject 3 | 0.5238 |
| Subject 4 | 0.6260 |
| Subject 5 | 0.6388 |
| Subject 6 | 0.5625 |
| Width of face = 0.68 cm (middle subject) | Euclidean distance in Figure 7a |
| Subject 1 | 0.5203 |
| Subject 2 | 0.5745 |
| Subject 3 | 0.3967 |
| Subject 4 | 0.3535 |
| Subject 5 | 0.5153 |
| Subject 6 | 0.4891 |
| Width of face = 0.39 cm (left subject) | Euclidean distance in Figure 7b |
| Subject 1 | 0.5937 |
| Subject 2 | 0.5409 |
| Subject 3 | 0.5835 |
| Subject 4 | 0.6728 |
| Subject 5 | 0.6107 |
| Subject 6 | 0.6113 |
| Width of face = 0.47 cm (middle subject) | Euclidean distance in Figure 7b |
| Subject 1 | 0.5723 |
| Subject 2 | 0.6466 |
| Subject 3 | 0.5223 |
| Subject 4 | 0.4681 |
| Subject 5 | 0.6230 |
| Subject 6 | 0.6096 |

## 3. Results

In this study, we developed a real-time AAS based on Internet of Things (IoT) technology and facial recognition. A Raspberry Pi 3B setup including a Raspberry Pi camera captures and then transfers six frames from each local video of a face to the cloud server. The development of the facial detection and recognition algorithms is based on the facilities available on the cloud server. In addition, in this study, we propose two strategies in particular: data serialization for storing facial features in SQLite easily and adaptive tolerance (with respect to the width of the facial image) for improving the performance of facial recognition using Euclidean distance.

### 3.1. The Performance of Adaptive Tolerance

This study proposes adaptive tolerance for improving the performance of face recognition using Euclidean distance. The tolerance for face recognition is not fixed, instead of Equation (2). In order to compare the performance of face recognition using fixed tolerance (Figure 7b) with adaptive tolerance (Figure 8), two different tolerances were

set in the face recognition using Euclidean distance. The student stood at the same location (Figures 7b and 8). If adaptive tolerance was adopted to set the tolerance in the face recognition using Euclidean distance, the student could be recognized.

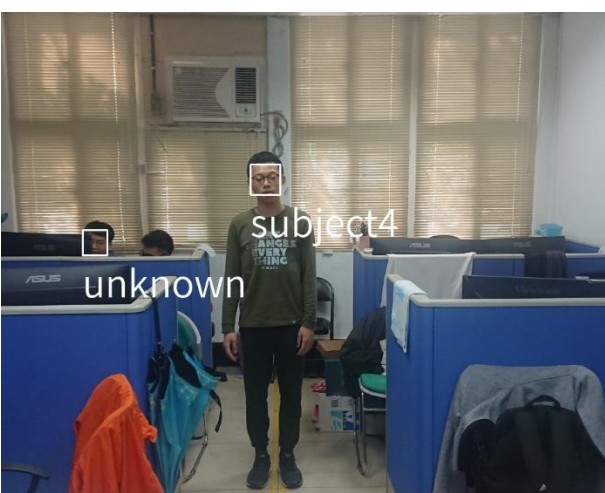

**Figure 8.** Adaptive tolerance for Euclidean distance in the facial recognition task.

### 3.2. The Performance of Face Detection Using the Dlib CNN and Dlib HOG Libraries

The Dlib library is arguably one of the most widely utilized packages for the purposes of facial recognition. This study compares the performance of face detection achieved by Dlib CNN with that achieved by Dlib HOG [42] in Figure 9.

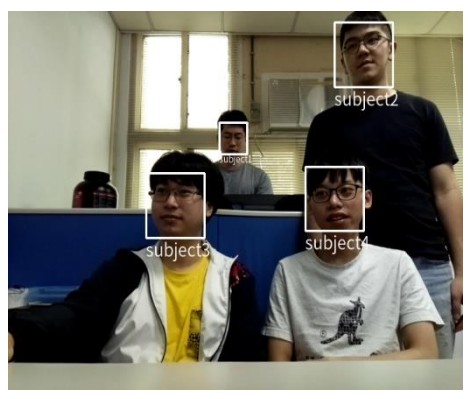 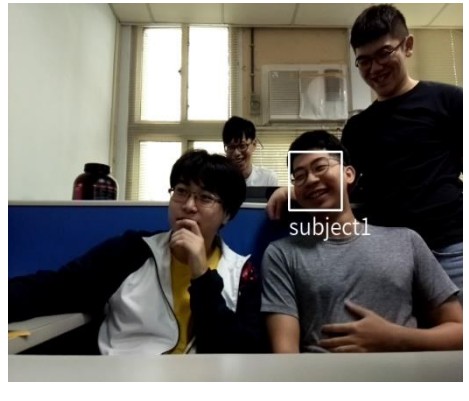

(**a**)                                          (**b**)

**Figure 9.** The performance of face detection using Dlib CNN (**a**) and Dlib HOG (**b**).

1.  Dlib CNN: A Maximum-Margin Object Detector (MMOD) CNN face detector that is highly accurate, very robust, and capable of detecting faces from varying viewing angles, in various lighting conditions, and withocclusions.
2.  Dlib HOG: A HOG + Linear SVM face detector that is accurate and computationally efficient.

Figure 9 shows that the Dlib CNN can detect the four faces presented within the object, but that Dlib HOG can only detect one face. Thus, Dlib CNN can be said to have better performance, in terms of detecting faces, than Dlib HOG. Hence, this study adopted the Dlib CNN library for the purpose of detecting faces and the positions of faces within frames.

### 3.3. Mask and Light Tests for Face Detection and Facial Recognition

The nature and competence of the extraction of facial features affect the performance of face detection and face recognition. The important facial features include: the location

of the eyes, and the relationship between the facial features and the overall contour of the faces. The ears and hairstyles are not determinants in terms of facial recognition because hairstyles are changeable, and so are often controlled or excluded, and the ears are almost always covered, which makes them less meaningful. In this study, we implemented our proposed facial detection and recognition methods, working under differing masking and lighting conditions, as shown in Figure 10a–i. Figure 10a–d indicates which conditions cause face detection and facial recognition failures, and Figure 10e–i indicates under which conditions successful face detection and recognition can be achieved.

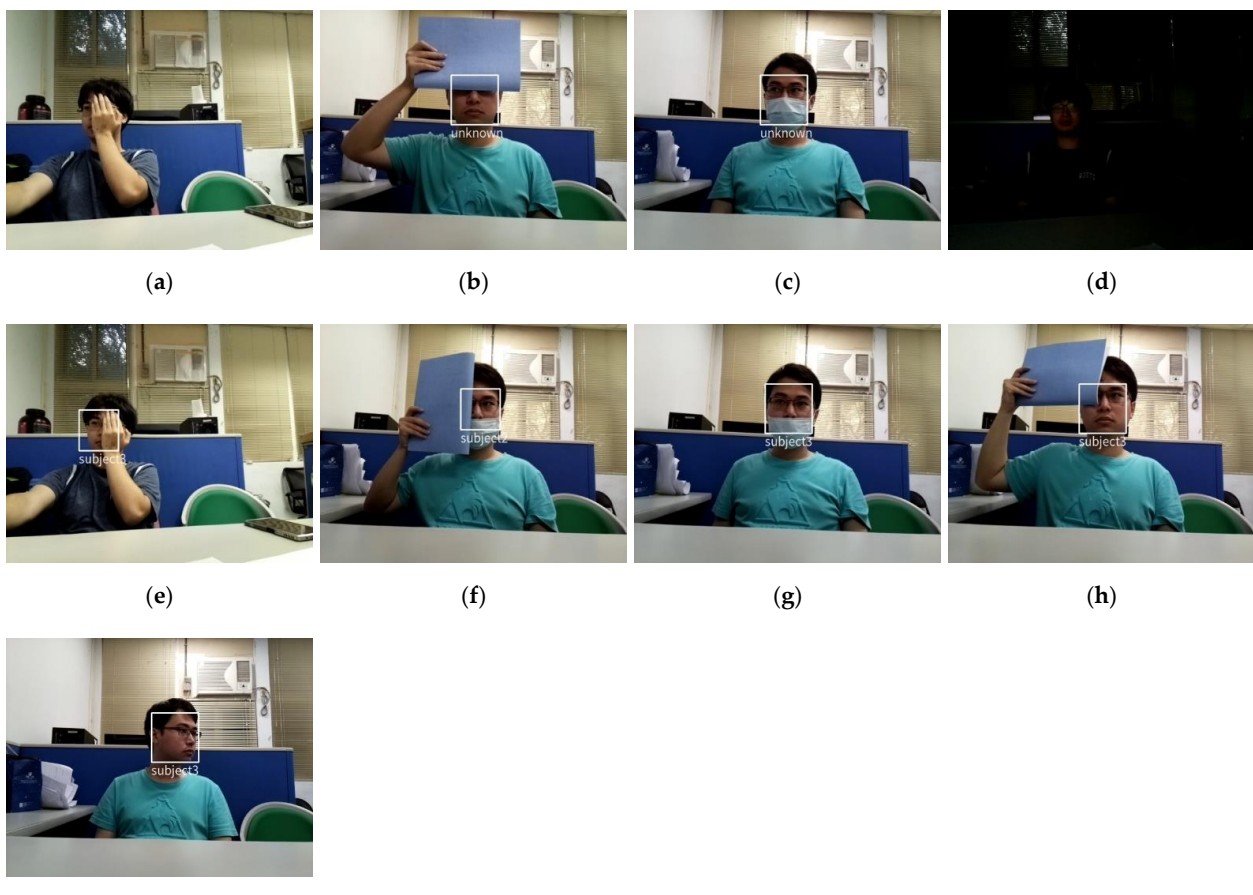

**Figure 10.** The performance of the proposed face detection and facial recognition methods under different masking and lighting conditions. (**a**) More than half of the facial features are covered. (**b**) Two eyes covered by a book. (**c**) Mouth and nose cover by a mask. (**d**) Environmental brightness is low. (**e**) Half of the facial features are covered. (**f**) Mouth and eye cover by a mask and book. (**g**) Mouth covered by mask. (**h**) One eye covered by a book. (**i**) Side profile.

According to the results shown in Figure 10, even when only half the facial features are visible, the system works; however, when the environmental brightness is low or more than half of the facial features are covered, face detection does not work. If most of the face is covered, very little information can be extracted by the Dlib library, and so the subsequent processing cannot work. Additionally, most face detection methods are significantly affected by changes in environmental illumination levels [43]. This proposed system will be set up in the classroom in order to register the attendance of students, and so it is unlikely that the environmental illumination levels will always be low enough to affect the facial recognition system significantly.

### 3.4. The AAS System

In order to test the performance of our AAS system (Figure 1), we set it up in the laboratory. The Raspberry Pi camera built into the Raspberry Pi 3B was thus able to capture relevant images, as shown in Figure 11. The Raspberry Pi transferred the images to the cloud server using an appropriate API. The results of the face detection and facial recognition processes are shown in Figure 9a. The face detection and facial recognition system implemented in the cloud server using the OpenCV and Dlib libraries successfully supported the automatic registration of the attendance of the students in the cloud server (Figure 12); the results were returned to the Raspberry Pi and shown on the computer screen.

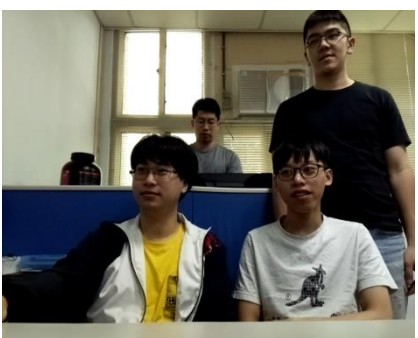

**Figure 11.** Example of an image that can be captured by the Raspberry Pi camera.

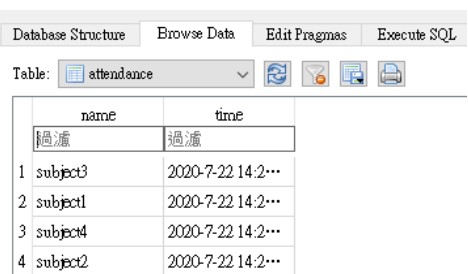

**Figure 12.** Roll call in SQLite which was sipped from the SQLite software results. The meaning of 過濾 is filter.

In addition, the AAS system was set up in the laboratory (Figure 11) and actually worked for one minute. The Raspberry Pi Model B sets 12 frames per second for the video stream, and extracts an image from every six frames of the live video. The AAS can work and record the attendance of the students every 0.5 s. Over the 120 occasions of four students' attendance the recording, the proposed system obtained 100% accuracy; i.e., four students could be detected and recognized in each image. When the AAS system with face detection using Dlib HOG and the adaptive tolerance for face recognition was involved, only one student could be recognized each time. When the AAS system with the face detection using Dlib CNN and with the fixed tolerance for face recognition was used, only three students could be recognized each time. The last student could be detected but not recognized. Thus, the results show that the proposed system is better than the AAS system at face detection using Dlib HOG, and the AAS system with face detection using Dlib CNN and a fixed tolerance for face recognition. Additionally, the performance of the proposed AAS system was compared with the previous study [44] which adopted Dlib HOG and Dlib CNN for face detection, deep residual networks (ResNets) for feature extraction, k-nearest neighbors (KNN) classifier for face recognition in the same video (Figure 11). The pervious study [44] obtained 99 % accuracy. Only a few images were not able to be used to detect and recognize all the four students.

## 4. Discussion

This study was a pilot study because we tested the performance of an AAS system in the laboratory rather than in the classroom. The study proposed a real-time AAS system schematic. Data serialization converted data objects (record images) present in complex data structures into a byte stream for storage, transfer, and distribution purposes on a physical device, which is a good tool for creating a dataset in SQLite. The results shown in Figure 8 reveal the adaptive tolerance for improving the performance of face recognition using Euclidean distance. In addition, Dlib CNN can be said to have a better performance in terms of detecting faces than Dlib HOG. Finally, this study tested a real-time AAS system in a laboratory setting, and the results shown in Section 3.4 indicate that the system was able to detect all of the students and could be registered in the cloud server automatically. In order to verify the performance of a real-time AAS system, we will apply to the Institutional Review Board (IRB) and test the AAS system in a classroom setting in our future work. In addition, this study adopted Euclidean distance with adaptive tolerance to face recognition. Future, the SVM and a more powerful classify method (CNN) will be adopted to compare its performance and efficacy to related face recognition methods.

## 5. Conclusions

In this study, we applied a combination of a number of common methods and libraries in order to develop an AAS system. In addition, we used our proposed data serialization and adaptive tolerance for Euclidean-distance methods to optimize the performance of this AAS. Data serialization was utilized in order to support the storing of facial features via SQLite. The calculation of adaptive (rather than fixed) tolerance was shown here to improve the performance of Euclidian-distance-based facial recognition. The results yielded by this study demonstrate that the proposed AAS can recognize multiple faces and so facilitate the automatic marking of attendance. Our proposed AAS would mean, in the classroom situation, that students would not be able to skip classes without the knowledge of their teachers.

Our development of this AI model and these techniques supports improvements in the design and application of facial recognition. However, the AAS implemented in this study also has some shortcomings which should be addressed—viz:

When the light levels in the classroom are low, this can mean that the proposed system manifests low accuracy in terms of face detection and recognition. In the future, we will explore methods by which the face detector system can be adjusted to work with low light levels.

**Author Contributions:** Conceptualization, C.-L.L. and Y.-H.H.; methodology, C.-L.L. and Y.-H.H.; software, Y.-H.H. validation, C.-L.L. and Y.-H.H.; formal analysis, Y.-H.H.; data curation, Y.-H.H.; writing—original draft preparation, C.-L.L. and Y.-H.H.; writing—review and editing, C.-L.L.; supervision, C.-L.L. All authors have read and agreed to the published version of the manuscript.

**Funding:** This research received no external funding.

**Conflicts of Interest:** The authors declare no conflict of interest.

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
