# Peer review of "The Application of Adaptive Tolerance and Serialized Facial Feature Extraction to Automatic Attendance Systems"

_electronics, doi:10.3390/electronics11142278_

Round 1

Reviewer 1 Report

The authors introduce a real-time automatic attendance system (AAS) based on the internet of things (IoT) and facial recognition. They also used a data serialization and an adaptive tolerance method that uses Euclidean distance to improve the performance of facial recognition processes.

Unfortunately, the paper contains the following problems.

The paper language needs polishing since it contains grammar and syntax errors.

There are reference error messages inside the text which need addressing (Error! Reference source not found).

At the end of the “Introduction” section, a small paragraph explaining the paper’s structure must be included.

All figures must contain a small paragraph explaining them. The paragraph must be placed right after the caption.

In lines 46-48, the authors claim that the use of RFID technology in the proposed system faces more security issues compared to biometric methods without mentioning those issues.

The literature review part at the end of the introduction section needs significant expansion with more papers regarding real-time facial recognition. The literature review must contain approximately 30 papers, and the authors must briefly explain the characteristics of each referenced paper.

The authors must include a separate subsection inside section 2 describing the structure of the Maximum-Margin Object Detector Convolutional Neural Network (MMOD-CNN). This section must contain the pseudocode of the MMOD-CNN method followed by a thorough explanation.

In the “Results” section, the authors did not present any metrics about the performance of their proposed system.

In the “Results” section, the authors did not present how their system performs compared to other existing methods.

The “Results” section lacks a table summarizing the parameters used in the conducted experiments.

A “Discussion” section must be included before the “Conclusion” section.

Author Response

  1. The paper language needs polishing since it contains grammar and syntax errors.

Reply: Thanks for your suggestions. We corrected the grammar and syntax errors carefully and also send the draft to Supreme Editing (https://www.supreme-editing.org/zh-tw/) which can help us to polish draft.

  1. There are reference error messages inside the text which need addressing (Error! Reference source not found).

Reply: I revised the reference error messages. The reason causes the reference error messages is error processing which transfer from Word file to pdf file.

  1. At the end of the “Introduction” section, a small paragraph explaining the paper’s structure must be included.

Reply: Thanks for your suggestions. We add the small paragraph explaining the paper’s structure.

“This study is organized as follows. In Section 2, the proposed AAS schematic and methods are described. Section 3 shows the results of this study. Section 4 discusses the proposed AAS system and indicates the advantage of the proposed methods. Section 5 provides concluding remarks.”

  1. All figures must contain a small paragraph explaining them. The paragraph must be placed right after the caption.

Reply: Thanks for your suggestions. We add a small paragraph explaining the figures.

  1. In lines 46-48, the authors claim that the use of RFID technology in the proposed system faces more security issues compared to biometric methods without mentioning those issues.

Reply: We add a paragraph to explain the security issues in RFID. The RFIS system is prone to manipulation. That is, another person can use same RFID to do the roll call.

“there are also some disadvantages, such as the fact that RFID is not as secure, again for this application, as biometric methods; the system is prone to manipulation[1]. That is, another person can use same RFID to do the roll call.”

  1. The literature review part at the end of the introduction section needs significant expansion with more papers regarding real-time facial recognition. The literature review must contain approximately 30 papers, and the authors must briefly explain the characteristics of each referenced paper.

Rely: Thanks for your suggestions. We add papers regarding papers regarding real-time facial recognition.

“Pandey et al., combined the on-demand resource availability of cloud computing, which can store and retrieve the captured video anytime. Also, it’s surveillance mechanism involves the Viola–Jones algorithm for face detection by analyzing captured data [2]. Shanthi and Svalakshmi developed a surveillance method by using a face recognition based Unmanned Aerial Vehicle (UAV) [3]. The method of UAV was implemented on Raspberry Pi module with Python libraries which included OpenCV, Dlib, Face_recognition and Numpy. Gupta and Singh used in-memory computation to develop a real-time face recognition system. Even the faces number is increasing, it can still keep frame rate during the entire process [4]. These recent studies developed the AAS system based on Raspberry Pi, cloud computing environment and face detection algorithms. They can make the systems simply, easily and more efficiently.”

We also explain some characteristics of each referenced paper.

“Parmar and Mehta described the most common methods of face recognition, such as the holistic matching method, the feature extraction method, and hybrid methods, and proposed a number of face recognition applications, such as face identification, access control, and security and identity verification [5]. Khandelwal et al., suggested various face recognition processing methods and adopted a convolutional neural network (CNN) to develop a face recognition algorithm [6]. Norouzi explored face recognition based on deep neural networks and described various facial expression recognition applications [7], such as student attendance systems, building security systems, etc. ”

“Kar et al., adopted the OpenCV and Light Tool Kit (FLTK) to develop an AAS system and suggested that OpenCV can provide a simple-to-use computer vision infrastructure that can help people build fairly sophisticated vision applications quickly. The OpenCV library contains over 500 functions that span many areas of vision [8]. Joseph et al., adopted face detection, 128-dimension face encoding extraction, and support-vector machine (SVM) training to develop a system [9]. Boyko et al., explored the features and analyzed the pros and cons of OpenCV and Dlib [10]. Xu et al., adopted Dlib to detect face then adopted CNN and a deep residual network (ResNet) for real-time face recognition [11]. Ambre et al., explored a real-time face recognition system using easily attainable components and libraries, such as Raspberry PI and Dlib, a Face Recognition library and OpenCV [12].”

  1. The authors must include a separate subsection inside section 2 describing the structure of the Maximum-Margin Object Detector Convolutional Neural Network (MMOD-CNN). This section must contain the pseudocode of the MMOD-CNN method followed by a thorough explanation.

Rely: Thanks for your suggestions. We add some description about MMOD-CNN in section2 and introduce how to use it.

“Dlib has two types of face detection tools (https://pyimagesearch.com/2021/04/19/face-detection-with-Dlib-hog-and-cnn/): one is based on a classic histogram of oriented gradient (HOG) features, in conjunction with linear classifier SVM, pyramid images, and a sliding window detection scheme; the other is based on Maximum-Margin Object Detector Convolutional Neural Network (MMOD-CNN), a face detector that uses deep learning face detection. The MMOD-CNN can be adopted using cnn_face_detection.py, with the function returning a list of rectangle objects which face within the object.”.

  1. In the “Results” section, the authors did not present any metrics about the performance of their proposed system.

Reply: Thanks for your suggestions. We show some metrics about the performance of the proposed system

“In addition, the AAS system was set up in the lab (Figure 11) and actually works for one minute. The Raspberry Pi Model B sets 12 frames per second for the video stream, and extracts an image from every six frames of the live video. The AAS can work and record the attendance of the students every 0.5 second. Over the 120 times of four students’ attendance recording, the proposed system can obtain 100% accuracy, i.e., four students can be detecting and recognized in each image. If AAS system with the face detection using Dlib HOG and the adaptive tolerance for face recognition is involved. Only one student can be recognized each time. If AAS system with the face detection using Dlib CNN and with the fixed tolerance for face recognition. Only three students can be recognition each time. Last student can be detected but be recognized. Thus, the results show that the proposed system is better than AAS system with face detection using Dlib HOG, AAS system with face detection using Dlib CNN and with the fixed tolerance for face recognition.

Figure 11 Example of an image captured by the Raspberry Pi camera

  1. In the “Results” section, the authors did not present how their system performs compared to other existing methods.

Rely: Thanks for your suggestions. We compared the performance of the proposed AAS system with the pervious study.

“The performance of the proposed AAS system is compared with the pervious study [13] which adopted Dlib HOG or Dlib CNN for face detect, Deep Residual Networks (ResNets) for feature extraction, k-nearest neighbors (KNN) classifier for face recognition in the same video (Figure 11). The pervious study [13] obtains 99 % accuracy. Only a few images were missing for detect and recognize all the four students.”

  1. The “Results” section lacks a table summarizing the parameters used in the conducted experiments.

Rely: This study adopts the python library to detect and recognize the face based on default parameters. Only a adaptive tolerance for face recognition using Euclidean distance is proposed in this study.

  1. A “Discussion” section must be included before the “Conclusion” section.

Rely: Thanks for your suggestions. We add “Discussion” section.

References

[1]       M. D. S. Solanke, "RFID TECHNOLOGY IN LIBRARIES," 2021.

[2]       S. Pandey, V. Chouhan, R. P. Mahapatra, D. Chhettri, and H. Sharma, "Real-Time Safety and Surveillance System Using Facial Recognition Mechanism," in Intelligent Computing and Applications: Springer, 2021, pp. 497-506.

[3]       K. Shanthi and P. Sivalakshmi, "Smart drone with real time face recognition," Materials Today: Proceedings, 2021.

[4]       N. K. Gupta and G. Singh, "In-Memory Computation for Real-Time Face Recognition," in Intelligent Computing and Applications: Springer, 2021, pp. 531-539.

[5]       D. N. Parmar and B. B. Mehta, "Face recognition methods & applications," arXiv preprint arXiv:1403.0485, 2014.

[6]       V. Khandelwal, V. Verma, and P. R. Devi, "Face Recognition Security System," EasyChair, 2516-2314, 2022.

[7]       M. Norouzi, "A Survey on Face Recognition Based on Deep Neural Networks," 2022.

[8]       N. Kar, M. K. Debbarma, A. Saha, and D. R. Pal, "Study of implementing automated attendance system using face recognition technique," International Journal of computer and communication engineering, vol. 1, no. 2, p. 100, 2012.

[9]       D. Joseph, M. Mathew, T. Mathew, V. Vasappan, and B. S. Mony, "Automatic Attendance System using Face Recognition," International Journal for Research in Applied Science and Engineering Technology, vol. 8, pp. 769-773, 2020.

[10]     N. Boyko, O. Basystiuk, and N. Shakhovska, "Performance evaluation and comparison of software for face recognition, based on dlib and opencv library," in 2018 IEEE Second International Conference on Data Stream Mining & Processing (DSMP), 2018: IEEE, pp. 478-482.

[11]      M. Xu, D. Chen, and G. Zhou, "Real-Time Face Recognition Based on Dlib," in Innovative Computing: Springer, 2020, pp. 1451-1459.

[12]     S. Ambre, M. Masurekar, and S. Gaikwad, "Face recognition using raspberry pi," in Modern Approaches in Machine Learning and Cognitive Science: A Walkthrough: Springer, 2020, pp. 1-11.

[13]     A. K. Tammisetti, K. S. Nalamalapu, S. Nagella, K. Shaik, and K. A. Shaik, "Deep Residual Learning based Attendance Monitoring System," in 2022 8th International Conference on Advanced Computing and Communication Systems (ICACCS), 2022, vol. 1: IEEE, pp. 1089-1093.

Reviewer 2 Report

This manuscript electronics-1760831 proposed a real-time automatic attendance system (AAS) based on the internet of things (IoT) and facial recognition. A Raspberry Pi Camera build in Raspberry Pi 3B is used to transfer the images to a cloud server. Face detection and recognition libraries are implemented in a cloud server which handles all the processes. In addition, this study proposes the application of data serialization processing and adaptive tolerance vis-a-vis Euclidean distance. The facial features encountered are processed with data serialization before they are saved in SQLite; this can easily be written and read back from the database. Examining the differences between the facial features already stored in the SQLite databases and any new facial features, the proposed adaptive tolerance system can improve the performance of the facial recognition method using Euclidean distance. My overall impression of this paper is that it is in general well-organized. It was a pleasure reviewing this work and I can recommend it for publication in Electronics after a major revision. I respectfully refer the authors to my comments below.

1.       The English needs to be revised throughout. The authors should pay attention to the spelling and grammar throughout this work. I would only respectfully recommend that the authors perform this revision or seek the help of someone who can aid the authors.

2.       In the Introduction part, “main contributions” is best to list clearly by breaking it down into three points. Pleases adjust the major contributions.

3.       The original statement “Thus, facial recognition [1] has attracted a great deal of research attention and has been subject to sustained development over the past 30 years; it is of great potential across numerous government and commercial applications [2][3-4].” ([1] https://doi.org/10.1016/j.infrared.2020.103594, [2] https://doi.org/10.1016/j.neucom.2020.05.081 [3] https://doi.org/10.1016/j.neucom.2020.09.068, [4] https://doi.org/10.1016/j.infrared.2019.103061)

4.       (Section 2) There is several mistake that “Error! Reference source not found.” Please correct this.

5.       (Section 1. Introduction) The reviewer suggests authors don't list a lot of related tasks directly. It is better to select some representative and related literature or models to introduce with certain logic. For example, the latter model is an improvement on one aspect of the former model.

6.       The sentence is revised as “However, with the development of information technology [][][][], this has been widely used to provide convenience, speed the task..”(DOI: 10.1109/TII.2021.3128240; https://doi.org/10.1016/j.neucom.2021.03.122; DOI: 10.1109/TII.2019.2934728; DOI: 10.1109/TMM.2021.3081873)

7.       (Section 3, Results) The reviewer suggests to add some compared methods to prove the performance of the proposed method.

8.       (Page 6, Section 2. Method) In Section 2, the reviewer suggests authors add some formal descriptions of the proposed methodology (in Fig. 6), so that the reader can better understand the process.

9.       (Page 2, Section I Introduction) Please add some references. The original statement is revised as “Although facial recognition based on deep learning [1][2] [3] techniques has high accuracy, the model is complex ..” ([1] https://doi.org/10.1016/j.infrared.2021.103823 [2] https://doi.org/10.1016/j.infrared.2022.104146, [3] DOI: 10.1109/TII.2022.3143605)

10.   The authors are suggested to add some experiments with the methods proposed in other literatures, then compare these results with yours, rather than just comparing the methods proposed by yourself on different models, such as “Facial expression recognition method with multi-label distribution learning for non-verbal behavior understanding in the classroom”.

11.   (Section 4. RESULTs) The reviewer suggest authors add more scenarios to compare with the state-of-the-art methods.

12.   (Section 5 Conclusion) Please point out the future work in this part. 

My overall impression of this manuscript is that it is in general well-organized. The work seems interesting and the technical contributions are solid. I would like to check the revised manuscript again.

Author Response

    1. The English needs to be revised throughout. The authors should pay attention to the spelling and grammar throughout this work. I would only respectfully recommend that the authors perform this revision or seek the help of someone who can aid the authors.

    Reply: Thanks for your suggestions. We corrected the grammar and syntax errors carefully and also send the draft to Supreme Editing (https://www.supreme-editing.org/zh-tw/) which can help us to polish draft.

    1. In the Introduction part, “main contributions” is best to list clearly by breaking it down into three points. Pleases adjust the major contributions.

    Reply: Thanks for your suggestions. We adjust the major contributions in the paper.

    “The aim of this study is to develop a real-time AAS system based on internet of things (IoT) technology and facial recognition which can handle the registration of students. The main contributions of this study are threefold:

    1. Building a cloud server to develop face detection and recognition algorithms. A Raspberry Pi Camera system built into a Raspberry Pi 3B will be used to capture and transfer images to a cloud server with a high-speed GPU. The development of our face detection and recognition algorithms will be based on this cloud server, which will handle the processes involved in registering student attendance, namely comparing faces using extracted features stored in a database.
    2. Creating an SQLite database to save students’ facial features based on data serialization, which we propose as the strategy for this process. In order to create the SQLite database, the students’ facial features will be put through data serialization before they are saved using SQLite.
    3. Improving the performance of face recognition using adaptive tolerance w.r.t. Euclidean distance. In comparing the facial features found on the database with the new facial features encountered in the input, the former needs to be subjected to data deserialization. The proposed adaptive tolerance method can improve the performance of facial recognition processes using Euclidean distance.”
    4. The original statement “Thus, facial recognition [1] has attracted a great deal of research attention and has been subject to sustained development over the past 30 years; it is of great potential across numerous government and commercial applications [2][3-4].” ([1] https://doi.org/10.1016/j.infrared.2020.103594, [2] https://doi.org/10.1016/j.neucom.2020.05.081 [3] https://doi.org/10.1016/j.neucom.2020.09.068, [4] https://doi.org/10.1016/j.infrared.2019.103061)

    Reply: Thanks for your suggestions. We cited these papers.

    1. (Section 2) There is several mistakes that “Error! Reference source not found.” Please correct this.

    Reply: I revised the reference error messages. The reason causes the reference error messages is error processing which transfer from Word file to pdf file.

    1. (Section 1. Introduction) The reviewer suggests authors don't list a lot of related tasks directly. It is better to select some representative and related literature or models to introduce with certain logic. For example, the latter model is an improvement on one aspect of the former model.

    Reply: Thanks for your suggestions. We explain some characteristics of each referenced paper.

    “Parmar and Mehta described the most common methods of face recognition, such as the holistic matching method, the feature extraction method, and hybrid methods, and proposed a number of face recognition applications, such as face identification, access control, and security and identity verification [1]. Khandelwal et al., suggested various face recognition processing methods and adopted a convolutional neural network (CNN) to develop a face recognition algorithm [2]. Norouzi explored face recognition based on deep neural networks and described various facial expression recognition applications [3], such as student attendance systems, building security systems, etc.”

    “Kar et al., adopted the OpenCV and Light Tool Kit (FLTK) to develop an AAS system and suggested that OpenCV can provide a simple-to-use computer vision infrastructure that can help people build fairly sophisticated vision applications quickly. The OpenCV library contains over 500 functions that span many areas of vision [4]. Joseph et al., adopted face detection, 128-dimension face encoding extraction, and support-vector machine (SVM) training to develop a system [5]. Boyko et al., explored the features and analyzed the pros and cons of OpenCV and Dlib [6]. Xu et al., adopted Dlib to detect face then adopted CNN and a deep residual network (ResNet) for real-time face recognition [7]. Ambre et al., explored a real-time face recognition system using easily attainable components and libraries, such as Raspberry PI and Dlib, a Face Recognition library and OpenCV [8].”

    “Pandey et al., combined the on-demand resource availability of cloud computing, which can store and retrieve the captured video anytime. Also, it’s surveillance mechanism involves the Viola–Jones algorithm for face detection by analyzing captured data [9]. Shanthi and Svalakshmi developed a surveillance method by using a face recognition based Unmanned Aerial Vehicle (UAV) [10]. The method of UAV was implemented on Raspberry Pi module with Python libraries which included OpenCV, Dlib, Face_recognition and Numpy. Gupta and Singh used in-memory computation to develop a real-time face recognition system. Even the faces number is increasing, it can still keep frame rate during the entire process [11]. These recent studies developed the AAS system based on Raspberry Pi, cloud computing environment and face detection algorithms. They can make the systems simply, easily and more efficiently.”

    1. The sentence is revised as “However, with the development of information technology [][][][], this has been widely used to provide convenience, speed the task..”(DOI: 10.1109/TII.2021.3128240; https://doi.org/10.1016/j.neucom.2021.03.122; DOI: 10.1109/TII.2019.2934728; DOI: 10.1109/TMM.2021.3081873)

    Reply: Thanks for your suggestions. We cited these papers.

    1. (Section 3, Results) The reviewer suggests to add some compared methods to prove the performance of the proposed method.

    Rely: Thanks for your suggestions. We show some metrics about the performance of the proposed system

    “In addition, the AAS system was set up in the lab (Figure 11) and actually works for one minute. The Raspberry Pi Model B sets 12 frames per second for the video stream, and extracts an image from every six frames of the live video. The AAS can work and record the attendance of the students every 0.5 second. Over the 120 times of four students’ attendance recording, the proposed system can obtain 100% accuracy, i.e., four students can be detecting and recognized in each image. If AAS system with the face detection using Dlib HOG and the adaptive tolerance for face recognition is involved. Only one student can be recognized each time. If AAS system with the face detection using Dlib CNN and with the fixed tolerance for face recognition. Only three students can be recognition each time. Last student can be detected but be recognized. Thus, the results show that the proposed system is better than AAS system with face detection using Dlib HOG, AAS system with face detection using Dlib CNN and with the fixed tolerance for face recognition.

    1. (Page 6, Section 2. Method) In Section 2, the reviewer suggests authors add some formal descriptions of the proposed methodology (in Fig. 6), so that the reader can better understand the process.

    Rely: Thanks for your suggestions. We add formal descriptions of the proposed methodology in Fig. 6.

    “Figure 6 shows the flowchart of the face recognition process proposed and described as follows:

    1. A real-time continuous face recognition AAS which captures the attendance and duration of attendance of students from the Raspberry Pi camera set-up requires the Raspberry Pi Model B.
    2. Raspberry Pi Model extracts an image from every six frames of the live video (of the classroom) and uploads this to the cloud server.
    3. The MMOD-CNN process has been adopted as the process by which the faces—and their positions within the object—are detected. The object is a list of rectangle objects. If the MMOD-CNN cannot obtain rectangle objects, we go back to Step 2.
    4. OpenCV crops the image of each face with the object and converts it to grayscale form.
    5. The estimated 128 facial landmarks are obtained using the Dlib facial recognition network.
    6. For facial similarity calculation, the estimated facial landmarks are compared with the facial landmarks currently stored in the SQLite database. This comparison is performed, after the JSON deserialization process has been applied to the existing facial features stored in the SQLite database, using Euclidean distance [12] as follows:

                           (1)

    where  is one of the facial landmarks which currently exists in the SQLite database and  is a just-estimated facial landmark derived from a new image of a student.

    1. If the value of the Euclidean distance is smaller than the tolerance, this means that the student whose image is being processed has been identified.
    2. The name of the student can be shown on the computer screen, and the student marked as present via the cloud server.”
    3. (Page 2, Section I Introduction) Please add some references. The original statement is revised as “Although facial recognition based on deep learning [1][2] [3] techniques has high accuracy, the model is complex ..” ([1] https://doi.org/10.1016/j.infrared.2021.103823 [2] https://doi.org/10.1016/j.infrared.2022.104146, [3] DOI: 10.1109/TII.2022.3143605)

    Reply: Thanks for your suggestions. We cited these papers.

    1. The authors are suggested to add some experiments with the methods proposed in other literatures, then compare these results with yours, rather than just comparing the methods proposed by yourself on different models, such as “Facial expression recognition method with multi-label distribution learning for non-verbal behavior understanding in the classroom”.

    Rely: This study was a pilot study because we tested the performance of an AAS system in the lab rather than in the classroom. In order to verify the performance of a real-time AAS system, we will apply to the Institutional Review Board (IRB) and test the AAS system in a classroom setting in our future work.

    1. (Section 4. RESULTs) The reviewer suggests authorsadd more scenarios to compare with the state-of-the-art methods.

    Rely: Thanks for your suggestions. We compared the performance of the proposed AAS system with the pervious study.

    “The performance of the proposed AAS system is compared with the pervious study [13] which adopted Dlib HOG or Dlib CNN for face detect, Deep Residual Networks (ResNets) for feature extraction, k-nearest neighbors (KNN) classifier for face recognition in the same video (Figure 11). The pervious study [13] obtains 99 % accuracy. Only a few images were missing for detect and recognize all the four students.”

    1. (Section 5 Conclusion) Please point out the future work in this part. 

    Rely: Thanks for your suggestions. We add future work in draft.

    “In order to verify the performance of a real-time AAS system, we will apply to the Institutional Review Board (IRB) and test the AAS system in a classroom setting in our future work. In addition, this study adopts Euclidean distance with adaptive tolerance to face recognition. Future, the SVM and more powerful classify method (CNN) will be adopted to comparing the performance and efficacy among related face recognition methods.”

    References

    [1]        D. N. Parmar and B. B. Mehta, "Face recognition methods & applications," arXiv preprint arXiv:1403.0485, 2014.

    [2]        V. Khandelwal, V. Verma, and P. R. Devi, "Face Recognition Security System," EasyChair, 2516-2314, 2022.

    [3]        M. Norouzi, "A Survey on Face Recognition Based on Deep Neural Networks," 2022.

    [4]        N. Kar, M. K. Debbarma, A. Saha, and D. R. Pal, "Study of implementing automated attendance system using face recognition technique," International Journal of computer and communication engineering, vol. 1, no. 2, p. 100, 2012.

    [5]        D. Joseph, M. Mathew, T. Mathew, V. Vasappan, and B. S. Mony, "Automatic Attendance System using Face Recognition," International Journal for Research in Applied Science and Engineering Technology, vol. 8, pp. 769-773, 2020.

    [6]        N. Boyko, O. Basystiuk, and N. Shakhovska, "Performance evaluation and comparison of software for face recognition, based on dlib and opencv library," in 2018 IEEE Second International Conference on Data Stream Mining & Processing (DSMP), 2018: IEEE, pp. 478-482.

    [7]        M. Xu, D. Chen, and G. Zhou, "Real-Time Face Recognition Based on Dlib," in Innovative Computing: Springer, 2020, pp. 1451-1459.

    [8]        S. Ambre, M. Masurekar, and S. Gaikwad, "Face recognition using raspberry pi," in Modern Approaches in Machine Learning and Cognitive Science: A Walkthrough: Springer, 2020, pp. 1-11.

    [9]        S. Pandey, V. Chouhan, R. P. Mahapatra, D. Chhettri, and H. Sharma, "Real-Time Safety and Surveillance System Using Facial Recognition Mechanism," in Intelligent Computing and Applications: Springer, 2021, pp. 497-506.

    [10]      K. Shanthi and P. Sivalakshmi, "Smart drone with real time face recognition," Materials Today: Proceedings, 2021.

    [11]      N. K. Gupta and G. Singh, "In-Memory Computation for Real-Time Face Recognition," in Intelligent Computing and Applications: Springer, 2021, pp. 531-539.

    [12]      H. Wu, Y. Cao, H. Wei, and Z. Tian, "Face recognition based on Haar like and Euclidean distance," in Journal of Physics: Conference Series, 2021, vol. 1813, no. 1: IOP Publishing, p. 012036.

    [13]      A. K. Tammisetti, K. S. Nalamalapu, S. Nagella, K. Shaik, and K. A. Shaik, "Deep Residual Learning based Attendance Monitoring System," in 2022 8th International Conference on Advanced Computing and Communication Systems (ICACCS), 2022, vol. 1: IEEE, pp. 1089-1093.

Round 2

Reviewer 1 Report

Reference errors still exist in the document (Error! Reference source not

found.)